# Clinical and Radiological Outcomes after Total Shoulder Arthroplasty Using Custom-Made Glenoid Components: A Systematic Review

**DOI:** 10.3390/jcm11247268

**Published:** 2022-12-07

**Authors:** Michael Stephan Gruber, Tamara Schwarz, Marlene Lindorfer, Felix Rittenschober, Martin Bischofreiter, Josef Hochreiter, Reinhold Ortmaier

**Affiliations:** 1Department of Orthopedic Surgery, Ordensklinikum Linz Barmherzige Schwestern, Vinzenzgruppe Center of Orthopedic Excellence, Teaching Hospital, Paracelsus Medical University Salzburg, 4020 Linz, Austria; 2Faculty of Medicine, Johannes Kepler University, 4020 Linz, Austria; 3Department of Orthopedic Surgery, Clinic Diakonissen Schladming, 8970 Schladming, Austria

**Keywords:** custom-made implant, shoulder arthroplasty, reverse total shoulder arthroplasty, glenoid bone loss, 3D printing

## Abstract

Reverse total shoulder arthroplasty presents itself sometimes as challenging when it comes to addressing massive bone loss, either in primary or revision settings. Custom components recently have made their way into shoulder prosthetics and are meant to help in the case of extensive glenoid bone destruction. Because of strict indication and the fairly recent introduction of these implants, the usage of custom-made glenoid implants is not very common yet. However, the early results are promising. The purpose of this review was to summarize and analyze the available literature. Therefore, a systematic review was performed according to PRISMA guidelines. A comprehensive search of the databases PubMed, Cochrane, and Livivo was performed to screen for studies reporting on clinical and radiological outcomes of custom glenoid implants. Four studies with a total of 46 shoulders were included in this review. The mean patient age was 68.8 years and the mean time of follow-up was 24.3 months. The weighted means showed an increase in CMS (32.7 points), in ASES (39.8 points), in anteversion (67.4 degrees), and in abduction (51.9 degrees) and a decrease in VAS (5.4 points). Custom-made glenoid implants are therefore a viable option in cases of large combined glenoid bone loss, both in primary and revision shoulder arthroplasty.

## 1. Introduction

In shoulder arthroplasty, it is all about restoring range of motion and ensuring stability of the shoulder joint. The introduction of Paul Grammont’s design of a reverse total shoulder prosthesis marked a milestone in these efforts. Since then, this type of implant has undergone several modifications and is used for various indications. Despite different possibilities such as glenoid bone grafting, metal augments, or eccentric reaming, the treatment of severe glenoid bone loss remains a challenging task. While overall numbers of implanted shoulder prostheses have been steadily increasing over the years [1], absolute numbers of complications in the case of primary or revision surgery have increased with them. Intraoperative or postoperative complications occur in up to 15% of primary surgery and in up to 40% of revision surgery [2].

Hence, difficult cases regarding glenoid bone loss in primary and revision surgery have gradually become an issue. The causes are partly problems with the implant itself (implant loosening, wear, or dislocation) or bone damage, while explantation of the glenoid implant in the case of revision and problems with soft tissue (instability) or preexisting medical conditions (infection, rheumatoid arthritis, congenital deformation, posttraumatic) result in damage of the glenoid bone [3,4,5]. The reconstruction of the glenoid bone and proper implant orientation and fixation are vital to the outcome of total shoulder arthroplasty [6]. Up until recently, eccentric reaming and bone grafting as well as conventional metal augments were the only options to address glenoid bone defects [7,8]. Since 2018, several studies reported the outcome of custom-made glenoid components in cases of primary and revision total shoulder arthroplasty. These are implants directly manufactured according to each unique bone defect. The planning is carried out by technicians and the surgeon based on a computed tomography scan, and the joint line is reconstructed with the help of a statistical shape-modeling concept [9]. The glenoid component itself consists of a metal base plate and a mesh structure filling the bone defect, and it comes with positioning and drilling guides. The advantages of such a prosthesis are maximum contact area with the bone, optimal screw position depending on bone density, and maximum length thanks to preoperative 3D planning, and accurate reconstruction of the native joint line based on statistical shape modeling.

Since this technique is new, clinical and radiological results are important to know whether the prosthesis delivers what it promises. Currently, there are several studies about this topic which examine small patient counts. This is because of strict indication and the fairly recent introduction of these implants. We have therefore made it our goal to summarize the existing literature in the form of a review.

This review was set out to help examine the effect of custom-made glenoid prostheses, evaluating studies in which these implants were investigated and thereby providing further scientific basis concerning the usage of these implants.

## 2. Materials and Methods

### 2.1. Searching Procedure and Eligibility Criteria

This review was carried out according to PRISMA (Preferred Reporting Items for Systematic Reviews and Meta-analyses) guidelines [10]. The protocol was registered at PROSPERO (Prospective Register Of Systematic Reviews; www.crd.york.ac.uk/PROSPERO; accessed on 1 March 2022) under the ID CRD42022311745 [11]. The review of the articles and documentation of the systematic review was conducted using CADIMA (https://www.cadima.info/index.php; accessed on 1 March 2022) [12,13].

A systematic review of the available literature available on the databases Livivo (Medline and others), Cochrane, and PubMed was performed using a combination of the terms “Custom”, “Custom-made”, “Patient specific” or “PSI” with “shoulder arthroplasty“ or “total shoulder arthroplasty”. The initial literature search was performed independently by two researchers (M.L. and T.S.), followed by a screening based on title and abstracts. The selection of the eligible studies was then carried out by two other researchers (M.G and F.R.) based on information given by full text.

Research reporting on minimum postoperative clinical and radiological outcomes of 12 months after total shoulder arthroplasty using custom-made glenoid components was included in this systematic review. The following eligibility criteria were applied: (1) study design of at least level IV according to Oxford Centre for Evidence-Based Medicine [14]; (2) a minimum follow-up time after surgery of 12 months; (3) studies written in English and German language, or other languages that could be translated using google translate or DeepL; (4) trials investigating patients with primary and revision surgeries using customized glenoid implants; (5) a report on the clinical outcome of at least two of the following: CMS (Constant Murley Score) [15], ASES (American Shoulder and Elbow Surgeons) Score [16], and abduction; (6) a reported radiological outcome equivalent to the count of radiological signs of implant loosening; and (7) a report on the number of complications and revision surgery.

No restrictions were applied regarding the date of publication, the patients’ age, pre-existing diseases or previous shoulder surgery. Studies still in pre-print stage were excluded.

### 2.2. Data Extraction

The collection of data included (1) demographic data, meaning sex, age at surgery, surgery side; (2) clinical data, meaning constant score, American Shoulder and Elbow Surgeons Score, Visual Analog Scale for pain, range of motion (active abduction and active anteversion); and (3) radiological and overall outcome, meaning count of loosening, complications, and revision surgery.

### 2.3. Statistical Analysis

Due to the inconsistent data of the included studies and the small population of each, heterogeneity could not be assured. Hence, a meta-analysis was not performed, but a qualitative analysis of the different studies was performed alongside with the calculation of weighted means. Statistical tests were not performed due to the reasons mentioned above.

## 3. Results

In total, 1128 studies were identified using the defined search terms (PubMed, *n* = 988, 26 March 2022; Cochrane, *n* = 20, 28 March 2022; Livivo, *n* = 120, 4 April 2022), and 1009 articles were included for screening after duplicate removal. During the title stage, 951 studies were eliminated, and after sorting out another 41 trials during the abstract check, 17 full-text articles were assessed. Four studies met the inclusion criteria and were included in the systematic review [17,18,19,20] (Figure 1). Two of these were retrospective multicenter studies, and two were retrospective single-center studies. Each study was carried out as a retrospective case series (Level IV according to Oxford Centre for Evidence-Based Medicine [14]). Three different implant designs were used (ProMade, LimaCorporate, Udine, Italy; Glenius glenoid reconstruction system, Materialise, Leuven, Belgium; VRS-Glenoid vault reconstruction system, Zimmer Biomet, Warsaw, IN, USA).

Altogether, 46 patients were included in the follow-up, of which 21 were male and 25 were female (Table 1). The studies reported a mean follow-up time between 18.22 months and 30.92 months. The mean age of the patients at the time of surgery was between 64 and 76.6 years. The detailed clinical outcome is summarized in Table 2.

### 3.1. Studies in Detail

The study by Bodendorfer et al. [20] performed a follow-up on 11 patients (12 shoulders, 1 patient underwent surgery on both shoulders) treated in three hospitals. They used the VRS system by Zimmer Biomet. Severe glenoid bone loss at or medial to the coracoid was the indication for surgery in every case (combined defect according to Antuna et al. [21] and Type III according to Page et al. [22]). All clinical measurements showed significant improvement. There was no significant difference in the outcome of primary surgery compared to revision surgery. The authors reported no signs for loosening and no complication at the time of last follow-up.

The study by Ortmaier et al. [17] performed a follow-up on 9 patients (10 shoulders) treated by a single surgeon in one hospital. They used the Glenius glenoid reconstruction system by Materialise in eight cases and the ProMade prostheses by LimaCorporate in two cases. They classified the bone loss according to Kocsis et al. [7], and reported four type 2 and six type 3 glenoid defects. All clinical measurements showed significant improvement. They compared the planned position and screws with the postoperative result in six cases with a CT-scan, which showed little difference in implant positioning (inclination 2.1 degrees, retroversion 3.4 degrees, posterior alignment 2.9 mm, superior alignment 0.9 mm, medial alignment 0.5 mm). The screws were correctly aligned and showed an accuracy of 95.4% of the planned to the actual intraosseous length. No complications or signs for loosening were reported.

The study by Porcellini et al. [19] conducted a follow-up on six patients treated in two hospitals. They used the custom-made implant ProMade by LimaCorporate. The bone loss was classified using the Antuna classification [21] (six combined bone defects) and the Seebauer classification (four cases E4, two cases C4) [23]. All patients showed improvement of the scores used. However, no information about significance of the improvement was provided. Two patients were found to have developed radiolucent lines in the X-rays, without progression over time. There was no revision, but one female patient faced a partial, non-traumatic dislocation of her shoulder joint.

The study by Rangarajan et al. [18] conducted a follow-up on 19 patients treated by a single surgeon in one hospital. Of these, 1 patient was lost at follow-up, and 18 completed the minimum follow-up of 1 year. They used the VRS system by Zimmer Biomet. The glenoid bone loss was classified according to Walch [24] (in case of glenohumeral arthritis, 7 patients; 1 A2, 4 B3, and 2 C), Favard [25] (in case of cuff tear arthropathy, 1 patient; E3) or Antuna [21] (in case of revision, 10 patients; all of them combined defects). All clinical scores improved significantly, as did the range of motion, except active external rotation (*p* = 0.06). The authors did not find any evidence for loosening. There were complications in four cases. One patient had an infection of the prosthesis and was treated in a different hospital and lost to follow-up. One patient suffered from instability and hematoma formation, which caused three further interventions with stabilizing measurements and wound revision. One patient required an allograft strut graft due to a cortical perforation of the humeral shaft. One patient suffered a fracture of the greater tuberosity during surgery.

### 3.2. Weighted Means Calculation

The weighted means of the pooled study population show an average age at the time of surgery of 68.8 years (range, 48–83) and a mean time of follow-up of 24.3 months (range, 12–52) (Table 1). The weighted means of the clinical follow-up show an increase of 32.7 points in CMS, 39.8 points in ASES score, 67.4 degrees in anteversion, and 51.9 degrees in abduction as well as less pain postoperatively (VAS decrease of 5.4 points) (Table 3).

**Table 2 jcm-11-07268-t002:** Summary of the clinical data of the included literature ^a^.

	Bodendorfer et al. [20]	Ortmaier et al. [17]	Porcellini et al. [19]	Rangarajan et al. [18]
CMS ^b^ (0–100)				
preOP	-	10.9 ± 4.2	15	24.6 ± 10.2
postop	-	51.7 ± 9.9	24.8	60.4 ± 14.5
ASES ^c^ (0–100)				
preOP	33	-	15.3	32 ± 18.2
postop	80	-	45.8	79 ± 15.6
VAS ^d^ (0–10)				
preOP	7	-	8	6.2 ± 2.9
postop	0	-	2.3	0.7 ± 1.3
Anteversion (deg)				
preOP	95	28 ± 10.3	41.7	53 ± 27
postop	150	132 ± 27	62	124 ± 23
Abduction (deg)				
preOP	-	19 ± 7.4	35.8	42 ± 17
postOP	-	121 ± 31.4	55	77 ± 15

^a^ Data are presented as mean ± standard deviation. ^b^ CMS, Constant Murley Score: 0–100 points [15]. ^c^ ASES, American Shoulder and Elbow Surgeons: 0–100 points [16]. ^d^ VAS, Visual Analog Scale for pain: 0–10 points [26].

## 4. Discussion

This systematic literature review was set out to examine the clinical and radiological outcomes after surgery using custom made glenoid components in total shoulder arthroplasty. A total of four studies were included after screening 1128 studies according to the inclusion criteria. Due to the small number of suitable studies and the small patient numbers, a meta-analysis on this topic was not performed. Alternatively, the results of the included studies are presented, compared, and discussed, and weighted means were calculated.

The main finding is that each study reported an improvement in all clinical measurements that were extracted from the trials (CMS, ASES, VAS, Anteversion, Abduction). The studies presented by Bodendorfer, Ortmaier, and Rangarajan showed significant increase in every score and in range of motion. Porcellini et al. reported only increases in scores and range of motion but did not cite statistical significance.

The weighted means showed an increase in CMS (32.7 points), in ASES (39.8 points), in active anteversion (67.4 degrees), and in active abduction (51.9 degrees) and a decrease in VAS (5.4 points). These values all exceed the minimal clinically important differences by far (CMS: 5.7; ASES: 13.6–27.1; VAS: 1.6–4.3; aAnte: 12°; aAbd: 7°) and are even well above the substantial clinical benefit (CMS: 19.1; ASES: 31.5; VAS: 3.2; aAnte: 35.4°; aAbd: 28.5°) [27,28,29]. However, Porcellini et al. did report on an increase in CMS and ASES of only 9.8 points and 30.5 points, respectively. In addition, the increase in range of motion was comparatively low (anteversion 20.3°, abduction 19.2°). Due to the relatively small patient cohort (six patients, mean follow-up time 31.7 months), these results barely affected the calculated weighted means. There were no differences in the patient population; if anything, the patient cohort of Porcellini et al. should be considered more appropriate because of its younger mean age compared with the pooled mean. The reason for the anomaly remains unknown.

Another important finding is that there were two complications during and three complications after surgery in 46 patients. This equals to 10.9% complication rate, leading to revision surgery in one case and loss to follow-up in another case due to treatment in a different facility (4.3% revision rate). The current literature covering CAD-CAM procedures states a complication rate of 24% (Chammaa et al., 2017, 37 patients, all primary procedures) [30]. In reverse total shoulder arthroplasty and anatomic total shoulder arthroplasty in revision arthroplasty, the complication rates are 22–24% and 28%, respectively [31,32]. However, it has to be noted that the majority of the registered complications in this review originate from one trial, while two of the four included trials stated no complications. In this study, four of nineteen patients were faced with a complication, which equaled to 21% complication rate and 10.5% revision rate. The question arises whether there were in fact more complications than usual in this particular study or whether the other studies did not report their complications or counted only revisions as complications. However, another possibility is that small sample size and short follow-up period led to biased results. Therefore, it is imperative to collect more data to address this question.

The evaluation of the radiological outcomes showed no glenoid loosening. This is described as a major issue for several other procedures treating glenoid bone loss, such as glenoid bone grafting [33,34]. Decreased contact area may be a considerable factor too [35]. Considering this, the presented result is quite remarkable. There are two important factors responsible for the excellent halt: (1) the planning of the screws according to bone quality ensures the best position, and (2) the custom implants are produced for the particular bone loss and therefore exactly matching the glenoid surface.

Due to the relatively new custom-made implants, there are only a few studies about this topic, and on top of that, they report on heterogeneous patient population as they often mix revision arthroplasty and primary arthroplasty in their studies. Only Ortmaier et al. investigated a relatively homogeneous population of 10 cases of revision arthroplasty treated in one hospital by one surgeon [17]. Rangarajan et al. investigated 19 cases of primary and revision arthroplasty treated by one surgeon in one hospital [18]. Bodendorfer et al. (twelve cases) and Porcellini et al. (six cases) investigated primary and revision arthroplasty, too, which were treated in different hospitals [19,20]. Furthermore, the bone defects were graded using different classifications and the clinical examination was done using several different scores. The radiographic follow-up was mostly done using standard X-rays in two planes, and CT-scans were performed in only one study to provide better quality of assessment.

A further limitation of this study is the low patient count. At the time of this review, the authors have no information about the total count of custom glenoid implants used so far.

The last point which is to be seen critically is that there were too few patients to perform statistical testing. Only weighted means were calculated, which provides an overview to some extent.

## 5. Conclusions

Custom made glenoid implants are a feasible option in cases of large combined glenoid bone loss, both in primary and revision shoulder arthroplasty. Compared to revision using traditionally glenoid augmentation techniques, they show a twofold lower complication rate.

The most important factor for further evaluation should be to agree and focus on consistent clinical investigation, better radiologic follow-up with postoperative CT scans, and to include more patients over a longer period of time. This should best be achieved in the context of multicenter studies.

## Figures and Tables

**Figure 1 jcm-11-07268-f001:**
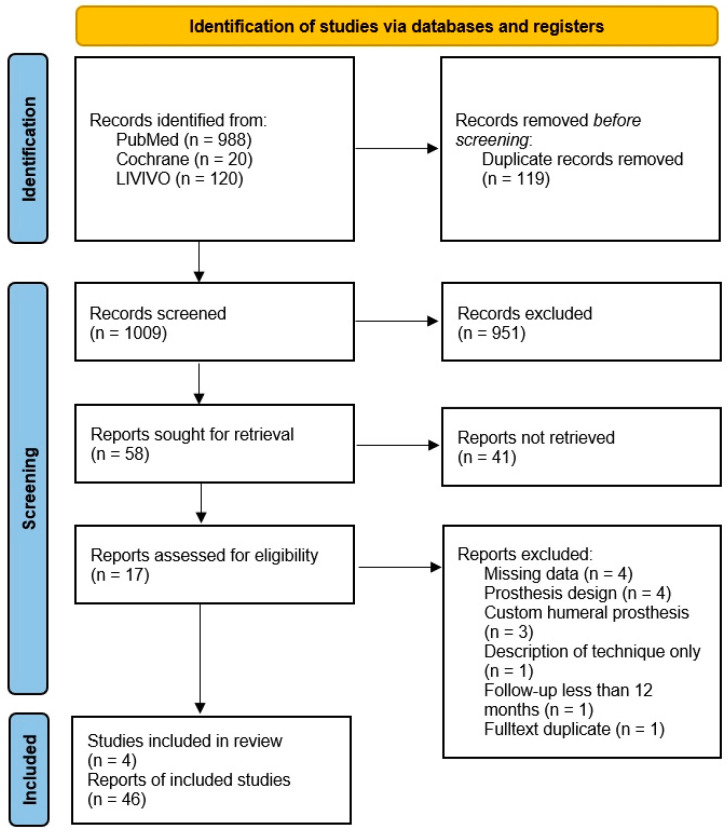
PRISMA Flow Diagram. Created according to the PRISMA guidelines [10].

**Table 1 jcm-11-07268-t001:** Studies reporting short-term outcome of custom glenoid implants—patient demographics of the included studies ^a^.

Author	Year	Shoulders	Primary/ Revision	Follow-Up (m)	Age (y)	Sex (m/f)
Bodendorfer et al. [20]	2020	12	7/5	30.9 (24–52)	68.0 (57–78)	7/5
Ortmaier et al. [17]	2022	10	0/10	23.1 (16–30)	76.6 (65–83)	0/10
Porcellini et al. [19]	2021	6	2/4	31.7 (25–38)	64.0 (48–74)	3/3
Rangarajan et al. [18]	2020	18	8/10	18.2 (12–27)	66.6 (50–80)	11/7
Summary		46	17/29	24.3 (12–52)	68.8 (48–83)	21/25

^a^ Data are presented as mean (range) unless otherwise indicated. y, years; m, months.

**Table 3 jcm-11-07268-t003:** Weighted means of the pooled data ^a^.

	preOP	postOP
CMS (*n* = 34) ^b^	18.9 (5–45)	51.6 (12–85)
ASES (*n* = 40) ^c^	26.6 (3.3–66.7)	66.4 (25–100)
VAS (*n* = 40) ^d^	6.1 (1–10)	0.7 (0–5)
Active Anteversion (*n* = 46)	57 (0–123)	124.4 (90–170)
Active Abduction (*n* = 34)	34.2 (0–70)	86.1 (45–170)

^a^ Data are presented as mean (range). ^b^ CMS, Constant Murley Score: 0–100 points [15]. ^c^ ASES, American Shoulder and Elbow Surgeons: 0–100 points [16]. ^d^ VAS, Visual Analog Scale for pain: 0–10 points [26].

## Data Availability

The data presented in this study are available on request from the corresponding author.

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
