# Peer review of "Clinical and Radiological Outcomes after Total Shoulder Arthroplasty Using Custom-Made Glenoid Components: A Systematic Review"

_jcm, 2022, doi:10.3390/jcm11247268_

Round 1

Reviewer 1 Report

Thank you for submitting your work. The authors reviewed and performed systematic review for postoperative outcomes for total shoulder arthroplasty using custom-made glenoid components. They followed the PRISMA guidelines. Four articles were included for final analysis. They did not perform the meta-analysis and statistical analysis due to the small number of suitable studies and small heterogenous patient population. Even though, they showed that custom-made glenoid implants seemed feasible option in case of large combined glenoid bone loss. Moreover, the complication rates were a nearly threefold less than previous reported outcomes in other applied options for glenoid bone loss, such as eccentric reaming, bone grafting, or metal augments.

 Overall, this is well organized, informative, and written article. However, it has some limitations, such as heterogeneity of the reverse implants and patient population, no control group, and small sample sizes/short-term follow-up, which seemed critical issue for arthroplasty. It seemed an early to proceed with such systematic review or meta-analysis because of newly designed and suggested implant. Moreover, more detailed strength points need to be addressed for persuading the surgeons to use this custom-made implant. Despite some limitations, this research provides interesting updates of recent literatures.  

Author Response

Thank you for your review.

We are aware of the fact that there is only little data on these new implants. To our mind, the high costs of custom-made prostheses (10.000-15.000€ per implant) and the little time since market introduction may explain this. With this review we hope to highlight the strengths of custom glenoid implants and encourage using them in difficult situations. In our understanding the price will decrease with rising implant numbers.

Reviewer 2 Report

Thank you for this systematic review. 

Even if the number of articles studied is very limited, such as the number of patients included or the short follow-up, the topic treated (reverse prosthesis in major bone loss) is very interesting. The introduction needs to be reworked (line 30 to 36). The discussion is a little short and also deserves to be reworked. Beyond the functional results, what is surprising, and must be analyzed in more detail, is the low ratio of complications. First : the instability, the first pitfall of this type of surgery where excessive medialization of the glenoid implant can increase the risk of this type of complication; Second : the absence of reported glenoid loosening. Indeed, in Pr Grammont's original concept, the center of rotation was at the contact between the bone and the metal. In this case, the center of rotation is very lateralized in the implant, which strongly modifies the constraints and the lever arm at the level of the implant/bone junction.  The very low complication rate remains remarkable. To be tempered with the short follow up.

Author Response

Thank you for reviewing our article. I focused on your comments and hope that they were addressed adequatly.

"The introduction needs to be reworked (line 30 to 36)":

In shoulder surgery, it is all about restoring range of motion and ensuring stability of the shoulder joint. The introduction of Paul Grammont’s design of a reverse total shoulder prosthesis marked a milestone in these efforts. Since then, this type of implant has undergone several modifications and is used for various indications. Despite different possibilities like glenoid bone grafting, metal augments or eccentric reaming, the treatment of severe glenoid bone loss remains a challenging task. While overall numbers of implanted shoulder prostheses have been steadily increasing over the years[1], absolute numbers of complications in the case of primary or revision surgery have increased with them. Intraoperative or postoperative complications occur in up to 15% of primary surgery and in up to 40% of revision surgery[2].

"Beyond the functional results, what is surprising, and must be analyzed in more detail, is the low ratio of complications."

Another important finding is that there were two complications during and three complications after surgery in 46 patients. This equals to 10.9 % complication rate, leading to revision-surgery in one case and loss to follow-up in another case due to treatment in a different facility (4.3 % revision rate). Current literature covering CAD-CAM procedures states a complication rate of 24% (Chammaa et al., 2017, 37 patients, all primary procedures)[30]. In reverse total shoulder arthroplasty and anatomic total shoulder arthroplasty in revision arthroplasty, the complication rates are 22-24% and 28%, respectively[31,32]. However, it has to be noted that the majority of the registered complications in this review originate from one trial, while two of the four included trials did state no complications. In this study, four of nineteen patients were faced with a complication, which equaled to 21% complication rate and 10.5 % revision rate. The question arises whether there were in fact more complications than usual in this particular study or whether the other studies did not report their complications or would have counted only revisions as complications. However, another possibility is that small sample size and short follow-up period led to biased results. Therefore, it is imperative to collect more data to address this question.

"... the absence of reported glenoid loosening. "

The evaluation of the radiological outcome showed no glenoid loosening. This is described as a major issue for several other procedures treating glenoid bone loss such as glenoid bone grafting11,17. Decreased contact area may be a considerable factor too10. Considering this, the presented result is quite remarkable. There are two important factors responsible for the excellent halt; 1) the planning of the screws according to bone quality ensures the best position, and 2) the custom implants are produced for the particular bone loss and therefore exactly matching the glenoid surface.